# Thrombospondin-1 Silencing Improves Lymphocyte Infiltration in Tumors and Response to Anti-PD-1 in Triple-Negative Breast Cancer

**DOI:** 10.3390/cancers13164059

**Published:** 2021-08-12

**Authors:** Elie Marcheteau, Thomas Farge, Michaël Pérès, Guillaume Labrousse, Julie Tenet, Stéphanie Delmas, Maud Chusseau, Raphaëlle Duprez-Paumier, Camille Franchet, Florence Dalenc, Caroline Imbert, Justine Noujarède, Céline Colacios, Hervé Prats, Florence Cabon, Bruno Ségui

**Affiliations:** 1Centre de Recherches en Cancérologie de Toulouse, INSERM UMR1037, CNRS UMR5071, 2 Aavenue Hubert Curien, CEDEX 1, 31047 Toulouse, France; emarcheteau@neovirtech.com (E.M.); thomas.farge@inserm.fr (T.F.); m.peres@chru-nancy.fr (M.P.); guillaume.labrousse@inserm.fr (G.L.); julie.tenet@univ-tlse.fr (J.T.); caroline.imbert@inserm.fr (C.I.); justine.noujarede@bric.ku.dk (J.N.); celine.colacios@inserm.fr (C.C.); herve.prats@inserm.fr (H.P.); florence.cabon@inserm.fr (F.C.); 2SeleXel, 1 Place Pierre Potier, BP 50624, CEDEX 1, 31106 Toulouse, France; stephanie.delmas@evotec.com (S.D.); maud.chusseau@imavita.com (M.C.); 3Université Toulouse III—Paul Sabatier, 118 Rte de Narbonne, 31062 Toulouse, France; 4Institut Claudius Regaud, Institut Universitaire du Cancer de Toulouse-Oncopole, 1 Av. Irène Joliot-Curie, 31100 Toulouse, France; Duprez-Paumier.raphaelle@iuct-oncopole.fr (R.D.-P.); franchet.camille@iuct-oncopole.fr (C.F.); dalenc.florence@iuct-oncopole.fr (F.D.); 5Equipe Labellisée par la Fondation ARC—Association Pour la Recherche sur le Cancer, 94803 Villejuif, France

**Keywords:** THBS1, TSP1, angiogenesis, metastasis, immunotherapy, tumor-infiltrating lymphocytes

## Abstract

**Simple Summary:**

Triple-negative breast cancer (TNBC) is associated with a poor prognosis, and the development of better therapeutic strategies is required. Herein, we investigated the role of the anti-angiogenic thrombospondin-1 (TSP1) in TNBC. TSP1 expression in tumor biopsies from TNBC patients was associated with a bad prognosis and a weak content of tumor-infiltrating lymphocytes (TILs). In the 4T1 mouse TNBC model, TSP1 knockdown reduced TGF-β activation and enhanced the content of TILs. Moreover, TSP1 knockdown decreased lung metastasis in syngeneic Balb/c immunocompetent mice but not in immunodeficient nude mice. Finally, TSP1 knockdown enhanced anti-PD-1 immunotherapy efficacy. Thus, targeting TSP1 may be considered as a putative therapeutic strategy in TNBC in combination with immunotherapy.

**Abstract:**

Triple-negative breast cancer (TNBC) is notoriously aggressive with a high metastatic potential, and targeted therapies are lacking. Using transcriptomic and histologic analysis of TNBC samples, we found that a high expression of thrombospondin-1 (TSP1), a potent endogenous inhibitor of angiogenesis and an activator of latent transforming growth factor beta (TGF-β), is associated with (i) gene signatures of epithelial–mesenchymal transition and TGF-β signaling, (ii) metastasis and (iii) a reduced survival in TNBC patients. In contrast, in tumors expressing low levels of TSP1, gene signatures of interferon gamma (IFN-γ) signaling and lymphocyte activation were enriched. In TNBC biopsies, TSP1 expression inversely correlated with the CD8+ tumor-infiltrating lymphocytes (TILs) content. In the 4T1 metastatic mouse model of TNBC, TSP1 silencing did not affect primary tumor development but, strikingly, impaired metastasis in immunocompetent but not in immunodeficient nude mice. Moreover, TSP1 knockdown increased tumor vascularization and T lymphocyte infiltration and decreased TGF-β activation in immunocompetent mice. Noteworthy was the finding that TSP1 knockdown increased CD8+ TILs and their programmed cell death 1 (PD-1) expression and sensitized 4T1 tumors to anti-PD-1 therapy. TSP1 inhibition might thus represent an innovative targeted approach to impair TGF-β activation and breast cancer cell metastasis and improve lymphocyte infiltration in tumors, and immunotherapy efficacy in TNBC.

## 1. Introduction

Triple-negative breast cancer (TNBC) remains associated with a poor prognosis, and the development of better therapeutics represents a major unmet clinical need. In metastatic settings, chemotherapy still represents the mainstay of treatment. Advances in genomics and molecular profiling have helped better define subtypes of TNBC with distinct biologic drivers to guide the therapeutic development of targeted therapies, including vascular endothelial growth factor A (VEGF-A), a major pro-angiogenic growth factor [1], immune checkpoint inhibitors for programmed cell death ligand 1 (PD-L1)-positive TNBC [2] and poly (ADP-ribose) polymerase (PARP) inhibitors for breast cancer 1/2 (BRCA1/2) mutant TNBC [3,4]. Whereas a PARP inhibitor has recently been shown to improve the prognosis of BRCA1/2 mutant TNBC [5], the usefulness of checkpoint inhibitors of the PD-1/PD-L1 axis needs to be better defined.

Thrombospondin-1 (TSP1) is a major endogenous anti-angiogenic protein secreted by various cell types, including cancer cells [6]. TSP1 is frequently repressed in the early stages of tumor development, contributing to an increased tumor angiogenesis [7]. Intriguingly, TSP1 is overexpressed in advanced stages of many tumor types such as prostate, pancreas, glioblastoma, breast and gallbladder cancers and melanoma [8,9,10]. Moreover, whereas TSP1 is expressed at very low levels in the normal breast, a high TSP1 level is associated with a bad prognosis in breast cancer [11]. Accordingly, TSP1 levels are higher in the plasma of metastatic patients as compared to non-metastatic patients [12,13,14]. TSP1 has pleiotropic functions and is considered as a major activator of transforming growth factor beta (TGF-β), a potent immunosuppressive and pro-metastatic cytokine [15,16,17]. As a matter of fact, it has been shown that TSP1 modulates the immune response and metastasis of cancer cells [9,18]. To the best of our knowledge, whether TSP1-dependent metastasis relies on its capacity to modulate the immune response remains to be established.

Herein, we investigated the impact of TSP1 expression in tumor biopsies from TNBC patients on immune responses and prognosis. In the 4T1 mouse TNBC model, we next analyzed the consequence of TSP1 knockdown on TGF-β activation, immune response and metastasis, in syngeneic Balb/c immunocompetent mice and in immunodeficient nude mice. Finally, we evaluated the impact of TSP1 knockdown on anti-PD-1 immunotherapy efficacy.

## 2. Materials and Methods

### 2.1. In Silico Analyses

Kaplan–Meier plots were generated with Kaplan–Meier Plotter, an online public database combining 35 independent transcriptomic studies, totaling 5143 breast tumor samples (www.kmplot.com (accessed on 1 April 2019) [19]). This database was used to determine the relevance of TSP1 mRNA expression to the overall survival (OS). Patients’ clinical data were collected and implemented in the online tool, allowing specific analysis in TNBC patients. The hazard ratio (HR) and 95% confidence intervals, as well as log rank *p*, were calculated and displayed on the webpage.

The TCGA-BRCA [20] (The Cancer Genome Atlas Research Network, Breast cancer) RNA-seq database was downloaded from TCGA servers with patients’ clinical data to perform transcriptomic analysis on TNBC only samples (*n* = 122). Patients were then classified as a function of their TSP1 (*THBS1*, ENSG00000137801) mRNA expression to segregate low and high TSP1 patients according to the median. We performed gene set enrichment analysis (GSEA) on these two groups using the GSEA v2.0 tool, developed from a joint project of UC San Diego (La Jolla, CA, USA) and Broad Institute (Cambridge, MA, USA) [19,21] and curated gene signatures available on their website. Normalized enrichment scores (NESs), *p*-values and false discovery rates were added to the figures as displayed by the GSEA result file.

### 2.2. Patients

A total of 12 patients with primary breast cancer (TNBC) were included in this study between July 2015 and February 2018. Patients who received neoadjuvant chemotherapy were excluded. Clinical characteristics of patients and tumor pathological features were obtained from their medical reports and followed the standard procedures adopted by the Claudius Regaud Institute.

### 2.3. Immunohistochemistry

IHC analysis of patient samples was performed on formalin-fixed, paraffin-embedded sections of the initial tumor biopsies with the indicated antibodies. The percentage of stained tumor cells was assessed for each biomarker. Antibodies against CD8 (clone SP57, Roche diagnostic, Meylan, France) and TSP1 (clone MA5-11330, Thermo Scientific, Villebon sur Yvette, France) were used. Alternatively, the proportion of TILs was determined as previously recommended [22].

### 2.4. Cell Lines and Transduction

4T1, a 6-thioguanine-resistant cell line selected from the 410.4 tumor without mutagen treatment, was obtained from ATCC and tested to be mycoplasma-free. Cells were maintained in Dulbecco’s modified Eagle’s medium (DMEM) with high glucose supplemented with 10% FBS.

Lentiviral vector pLKO.1 expressing the non-target shRNA (SHC002) or shRNA construct targeting TSP1 exon 5 (TRCN0000039696) was purchased from Sigma-Aldrich (Saint-Quentin-Fallavier, France). Plasmids were transfected into cells in culture using the TransIT-X2 Dynamic Delivery System (MIRUS BIO). Lentiviral vector pTRIP (Dharmacon, Illkirch, France) expressing Luciferase Firefly was produced in HEK293FT cells by the tri-transfection method using calcium phosphate. Lentiviral particles were harvested 72 h after transfection. For lentiviral transduction, 5 × 10^4^ cells were seeded onto a 12-well plate and incubated for 6 h with lentivirus in OPTI-MEM.

### 2.5. Animal Model

Six-week-old female BALB/cOlaHsd mice were purchased from Envigo (Gannat, France). Experiments respected institutional guidelines and were approved by the Institutional Animal Care and Research Advisory Committee (CEEA-122). To assess tumor growth, 0.1 million cells expressing luciferase were injected into the Balb/c female mammary fat pad. Tumor growth was monitored twice a week using a caliper.

### 2.6. Quantification of Lung Metastasis

Evaluation of the metastatic load was performed by quantification of luciferase mRNAs in lungs by RT-qPCR and normalized Cyclophilin-A. Primers’ sequences were as follows:Luciferase forward primer: 5′TCTAAAACGGATTACCAGGGATTT;Luciferase reverse primer: 5′ACCGGGAGGTAGATGAGATGTG;Cyclophilin A forward primer: 5′GTCAACCCCACCGTGTTCTT;Cyclophilin A reverse primer: 5′CTGCTGTCTTTGGGACCTTGT.

### 2.7. Immunotherapy in Mice

A neutralizing anti-mouse PD-1 antibody was administered i.p. at 0.2 mg on days 4, 6, 9 and 11 after tumor graft. Anti-PD-1 antibody (clone RMP1-14) and isotype control antibodies (clone 2A3) were purchased from BioXcell (Supplied by Euromedex, Souffelweyersheim, France).

### 2.8. Flow Cytometry

On day 11, mice were sacrificed, and tumors were collected and digested with Mouse Tumor Dissociation kit and GentleMacs (Miltenyi, Paris, France). Cells were counted and incubated with anti-CD16/32 blocking antibodies prior to incubation. Cells were stained with LIVE/DEAD reactive dye (Invitrogen, Villebon sur Yvette, France), and cell surface staining was performed on tumor-infiltrating and draining lymph node cells using anti-CD45 (clone 30-F11), anti-Thy-1 (clone 30-H12), anti-CD4 (clone GK1.5), anti-CD8 (clone 53–6.7) or anti-PD-1 (clone J43) antibody. Intracellular staining of Foxp3 (anti-mouse Foxp3 clone FJK-16s) and Ki67 (anti-mouse Ki67 clone B56) was performed with fix perm solution (eBioscience, Villebon sur Yvette, France). Cells were analyzed with a BD LSR Fortessa X-20 (BD Biosciences, Le Pont-de-Claix, France), followed by Flow-jo software analysis.

### 2.9. Statistical Analysis

Statistical significance of difference between groups was evaluated using the Graph-Pad Prism software (Ritme, Paris, France). Results are expressed as mean ± SEM or range, as appropriate. The Mann–Whitney test was used to compare two groups. Comparison between tumor growth curves was performed using a two-way ANOVA test. Significance was assumed at *p* < 0.05.

## 3. Results

### 3.1. TSP1 Expression Is Associated with a Poor Prognosis in TNBC Patients and a Low CD8+ T Cell Infiltration in Tumors

We first evaluated the correlation between TSP1 expression and prognosis in tumor samples of patients affected with breast cancer. Analysis of transcriptomic data combined from 35 independent studies (www.kmplot.com (accessed on 1 April 2019) [19]) showed that the TSP1 mRNA level, encoded by the *THBS1* gene, was significantly associated with a reduced survival in breast cancers, including ER-/PR-/HER2-TNBC (Figure 1A). Of note, the hazard ratio (HR) was higher in TNBC (HR = 1.85) as compared to all breast cancers (HR = 1.19), indicating that high TSP1 levels may contribute to tumor aggressiveness in TNBC patients mainly. To obtain insight into the putative mechanisms associated with TSP1 expression and cancer progression, we evaluated gene expression pathways by Gene Set Enrichment Analysis (GSEA) of 122 tumor samples from the TCGA TNBC cohort. The “Epithelial Mesenchymal Transition” and “Hallmarks of the TGF-β signaling” pathways were significantly enriched in tumors expressing high TSP1 levels (Figure 1B and Appendix A), whereas low expression of TSP1 was associated with the “Lymphocyte activation” and “Hallmarks of the interferon gamma response” pathways (Figure 1C and Appendix A). Accordingly, expression of genes encoding IFNγ, CD8α and CD8β was significantly increased in tumor samples with a low TSP1 expression (Figure 1D). We next analyzed TSP1 expression and TILs in primary tumor samples from TNBC patients in a prospective cohort by immunohistochemistry. We found an inverse correlation between TSP1 expression and T cell tumor infiltration, including CD8+ TILs (Figure 2A,B).

Collectively, these data associate TSP1 expression in patients’ tumor samples with an invasive phenotype and immune escape mechanisms. To study the underlying mechanisms, we then used the 4T1 syngeneic mouse model of metastatic TNBC.

### 3.2. TSP1 Inhibition in Murine Breast Cancer Cells Thwarts Metastasis in Immunocompetent but Not Immunodeficient Mice

TSP1 expression in 4T1 murine breast cancer cells was knocked down by stable transfection of a TSP1-shRNA (Figure 3A). TSP1 knockdown did not affect the cell doubling time in vitro (Figure 3B). In vivo, the tumor take-rate and growth of the 4T1 control (4T1 shCTL) and TSP1 knocked-down (4T1 shTSP1) cells orthotopically implanted in the mammary fat pad were not significantly different in syngeneic immunocompetent Balb/c mice (Figure 3C). However, TSP1 knockdown in 4T1 cells reduced their lung metastatic diffusion in immunocompetent mice (Figure 3D). Under our experimental conditions, six out of nine wild-type mice (66%) grafted with 4T1 shCTL cells showed metastatic nodules on the lungs compared to two out of nine wild-type mice (22%) injected with 4T1 shTSP1 cells (Figure 3D). To quantify the total metastatic load, we measured the luciferase mRNA level by RT-qPCR (Figure 3E). The metastatic lung dissemination of 4T1shT cells was significantly reduced as compared to 4T1 shCTL cells (Figure 3E). To assess the involvement of the adaptive immune response in the modulation of tumor progression, we performed the same experiments in nude mice, which lack T cells. The primary tumor growth (Figure 3C) as well as lung metastases (Figure 3D,E) was enhanced in nude mice, as compared to immunocompetent mice. Strikingly, TSP1 knockdown did not compromise 4T1 lung metastasis in nude mice.

Collectively, these data demonstrate that, whereas TSP-1 is not mandatory for 4T1 primary tumor growth, it contributes to lung metastasis. Since TSP1 knockdown had no effect in nude mice, we hypothesized that TSP1 impairs T cell-dependent immune responses, facilitating 4T1 lung metastasis.

### 3.3. Impacts of TSP1 Knockdown on Tumor Vascularization and T Cell Infiltration

The most striking feature regarding TSP1 inhibition in 4T1 tumors was the significant increase in the density of intratumoral blood vessels, as evaluated by CD31 staining (Figure 4A,B). This increased vascularization, observed at day 11 on IHC cuts of tumors, appeared to be, at least partly, functional since the same samples showed a significant decrease in CAIX-positive hypoxic zones in 4T1 shTSP1 tumors (Figure 4A,B). In good agreement with our in silico analyses, TSP1 knockdown was also associated with a significant decrease in active TGF-β in tumors, and with a significant increase in CD8α staining (Figure 4A,B). As a matter of fact, TSP1 knockdown did not significantly enhance the tumor density of high endothelial venules (HEV), which are major gateways for lymphocyte entry in breast tumors (Appendix A) [23].

To further investigate the consequences of TSP1 knockdown on the immune response, we performed flow cytometry analyses of lymphocytes in the tumor bed (Figure 4C–E), and tumor-draining lymph nodes (Appendix A). These experiments confirmed the significant increase in the proportion of CD8+ T cells among the total cells in TSP1 knockdown tumors and showed similar trends for CD4+ T cells and regulatory T cells (Figure 4D). Of note, the CD8/CD4 and CD8/CD4 Foxp3 ratios remained unchanged (Figure 4E and Appendix A), likely reflecting a non-selective increase in lymphocyte tumor infiltration upon TSP1 knockdown. The augmentation of TILs in shTSP1 tumors was not associated with significant changes in the proliferation of T cells, as evaluated by KI67 staining (Appendix A). Whereas TSP1 knockdown in 4T1 cells did not alter the T cell content in tumor-draining lymph nodes, it significantly increased the CD8+ T cell/Treg ratio (Appendix A). The latter observation indicates that the effects of TSP1 knockdown in tumor cells on the immune response are unlikely restricted to the primary tumors, enhancing the immune response in tumor-draining lymph nodes.

Collectively, these data show that TSP1 knockdown increases functional tumor angiogenesis and lymphocyte tumor infiltration.

### 3.4. TSP1 Knockdown Enhances Anti-PD-1 Therapy Efficacy in Mice

Only a small proportion of patients affected with TNBC exhibit objective clinical responses upon targeting PD-1 or PD-L1 with blocking monoclonal antibodies [24,25]. Tumor vascular normalization has been shown to enhance the efficacy of anti-PD-1 therapy in mice [26]. We thus sought to evaluate a potential role of TSP1 in anti-PD-1 therapy. As immunotherapy-induced anti-tumor immune responses depend on the expression of PD-1 and its ligands (i.e., PD-L1/2) [27], we first analyzed the expression of the *PDCD1* (encoding PD-1), *PDCD1LG1* (encoding PD-L1) and *PDCD1LG2* (encoding PD-L2) genes in TNBC tumor samples expressing TSP1 at low or high levels from the TCGA database. High TSP1 expression was associated with a significant reduction in PD-1 levels. Whereas the expression of PD-L2 was similar in both groups, PD-L1 expression was slightly, yet not significantly, reduced in TNBC tumor samples exhibiting high levels of TSP-1 (Figure 5A).

We next analyzed the expression of PD-1 on CD8+ TILs from 4T1 shCTL and shTSP1 tumors in mice (Figure 5B). Whereas the proportion of PD-1+ cells among CD8+ TILs was similar in both groups (Figure 5C), the proportion of PD-1+ CD8+ TILs among total cells significantly increased upon TSP1 knockdown (Figure 5D), likely reflecting the increase in CD8+ T cell infiltration in 4T1 shTSP1 tumors (as shown in Figure 4). Moreover, PD-1 expression was significantly augmented on CD8+ TILs, as evaluated by the increase in the mean fluorescence intensity (Figure 5E). Thus, TSP1 knockdown was associated with a PD-1 expression increase at the cell surface of CD8+ TILs from 4T1 tumors in mice. Finally, we investigated the impact of TSP1 knockdown on anti-PD-1 therapy in immunocompetent mice (Figure 5F and Appendix A). We found that the anti-PD-1 therapy efficacy towards 4T1 tumors was significantly enhanced by TSP1 knockdown under our experimental conditions (Figure 5F).

Collectively, the data indicate that silencing TSP1 is associated with an upregulation of PD-1 on CD8+ TILs and enhances the efficacy of anti-PD-1 therapy in TNBC.

## 4. Discussion

Tumor angiogenesis plays a critical role in cancer progression and resistance to therapies. Normalization of tumor vessels is an interesting strategy for cancer treatment [28,29]. We provide evidence that TSP1, produced by breast cancer cells, not only impairs functional tumor angiogenesis but also likely contributes to immune escape mechanisms, leading to metastasis and resistance to anti-PD-1 therapy in TNBC.

Our analyses of public human breast cancer databases strongly indicate that a high expression of the TSP1 mRNA is associated with a bad prognosis in TNBC. The TSP1 mRNA level was significantly associated with a reduced survival in breast cancers, as depicted in Figure 1A. One should note, however, that in all breast cancers, the HR is rather low (HR = 1.19 (1.07–1.33)) between patients exhibiting high and low levels of TSP1 mRNA. In sharp contrast, the HR (HR = 1.85 (1.19–2.88)) was greater in TNBC, indicating that a high TSP1 mRNA expression level in tumor samples can be considered as an independent bad prognosis biomarker in TNBC, but not in all breast cancers. In agreement with this observation, a sustained expression of TSP1 is also associated with the most aggressive stages of renal, prostate and pancreatic cancers [9,30]. The dosage of TSP1 in plasma shows that a high level is associated with a ganglionic infringement in breast cancer [31]. In 2015, the team of Campone also associated TSP1 with the metastatic relapse of TNBC and proposed its use as a marker of bad prognosis [10]. Thus, with regard to our own results, and those from the literature, the expression of TSP1 is correlated with tumor aggressiveness in numerous cancers, including TNBC.

Whereas our pilot study based on IHC analyses is limited due to the low number of cases (i.e., 12 patients with TNBC), the inverse correlation between anti-cancer immune response and TSP1 expression is further documented by our in silico analyses on 253 TNBC patients. In TNBC, we were also able to highlight that the strong expression of TSP1 was associated with a gene signature of epithelial–mesenchymal transition (EMT) and TGF-β signaling, a potent inductor of EMT. In addition, gene signatures of lymphocyte activation and IFN-γ response were both enriched in TNBC specimens expressing low TSP1 levels. On biopsies of patient tumors, we observed that CD8+ T cell infiltration was statistically increased in tumors expressing low TSP1 levels. The set of our data suggests that TSP1 not only contributes to the activation of TGF-β and the EMT process but also to immune escape. TGF-β signaling, as evaluated by a TGF-β-dependent gene signature, is enriched in tumor samples exhibiting high TSP1 mRNA levels (see Figure 1B). However, TGF-β mRNA levels are similar in tumor samples exhibiting high or low TSP1 expression levels (see Figure 1D). Those data further reinforce the notion that TSP1 acts at the protein rather than at the mRNA level, in order to activate the latent forms of TGF-β.

Considering the functional interaction between TSP1 and CD47, we cannot rule out the idea that it contributes to stimulating tumor progression, by impairing tumor vasculature normalization and/or the immune response. Indeed, TSP1 drives human and mouse endothelial cell senescence in a CD47-dependent manner [32]. CD47 is also expressed on various cancer cell types, including breast cancer cells, and behaves as a “don’t eat me signal” [33]. Thus, its interaction with TSP1 could prevent tumor cell phagocytosis by antigen-presenting cells, thereby limiting the priming of the immune response towards tumor cells.

To clarify the role of TSP1 in vivo, we selected the 4T1 cell line, which is a TNBC mouse model with a strong metastatic potential. We did not observe any impact on the growth of the primary tumors by knocking down TSP1 in the tumor cells. However, TSP1 knockdown triggered a significant increase in the tumor blood vessel density. These vessels were also more functional since we noticed a significant decrease in hypoxia. It is thus interesting to note that the vascular normalization and better oxygenation of tumors do not inevitably lead to an increase in the primary tumor size in this 4T1 model. We quantified lung metastases, and we show that TSP1 inhibition by shRNA entails a significant reduction in the number of animals presenting this type of metastatic dissemination. Thus, we provide evidence that TSP1 expressed by 4T1 cells is pro-metastatic, as observed in other tumor types [9,12,34,35,36,37]. Noteworthy, we show that the anti-metastatic effect of TSP1 silencing in syngeneic mice is abolished in immunodeficient mice (i.e., nude mice), which strongly suggests that the TSP1 pro-metastatic effect in our preclinical TNBC model depends on its capacity to inhibit the adaptive immune response.

Our transcriptomic data analysis in human TNBC samples suggests that TSP1 plays an important role in the inhibition of lymphocyte-dependent immune responses, driving the tumor progression and aggressive phenotype of this type of cancer. On TNBC biopsies, we noticed a negative correlation between TSP1 protein expression and total TILs, as evaluated by morphological criteria [22], including CD8+ TILs. These TILs are associated with a good prognosis in breast cancers, including in TNBC, and a better response to chemotherapies compared with poorly infiltrated tumors [30,31,38].

To estimate the role of TSP1 on the immune response, we studied lymphocyte infiltration in primary tumors that developed in immunocompetent mice transplanted with 4T1 shCTL or 4T1 shTSP1 cells. We noticed a significant increase in the infiltration of CD8+, CD4+ and Foxp3+ T cells when TSP1 was knocked down. It should be noted that the CD8+/CD4+ T cells and CD8+ T cell/Treg ratios remained constant. We therefore highlight, for the first time, that the inhibition of TSP1 expression enhanced tumor infiltration by various T lymphocyte subpopulations, including not only CD8+ and CD4+ T cells but also Tregs. These results are in line with studies showing that high tumor infiltration with both CD8+ T cells and Tregs can be associated with a good prognosis in breast cancers [38,39,40]. Of note, considering that Foxp3 can be upregulated in conventional T cells upon activation in humans [41], unlike in mice, the infiltration of Tregs in human samples based on the sole analysis of Foxp3 expression may conduct to misinterpretation.

Furthermore, we noticed an increase in the expression of PD-1 on CD8+ T cells in 4T1shTSP1 tumors. This increase, observed at day 11, probably reflects a better activation state, rather than an exhausted phenomenon of lymphocytes. Indeed, the CD8+ TILs were KI67+, indicating their proliferative capacity. TSP1 inhibition allows rendering 4T1 tumors significantly more sensitive to anti-PD-1 treatment. This effect could be connected to the increase in tumor vascularization, allowing a better intratumoral diffusion of anti-PD-1. In addition, the increase in target cells in tumors (i.e., CD8+ PD-1+ TILs) constitutes a predictive element of the response to anti-PD-1 [42].

## 5. Conclusions

This proof-of-concept study indicates that targeting TSP1 may represent a good strategy in TNBC to facilitate the anti-tumor immune response. Moreover, TSP1 could be considered as a bad prognosis biomarker in patients affected with TNBC and as a therapeutic target in combination with anti-PD-1 blocking antibody.

## Figures and Tables

**Figure 1 cancers-13-04059-f001:**
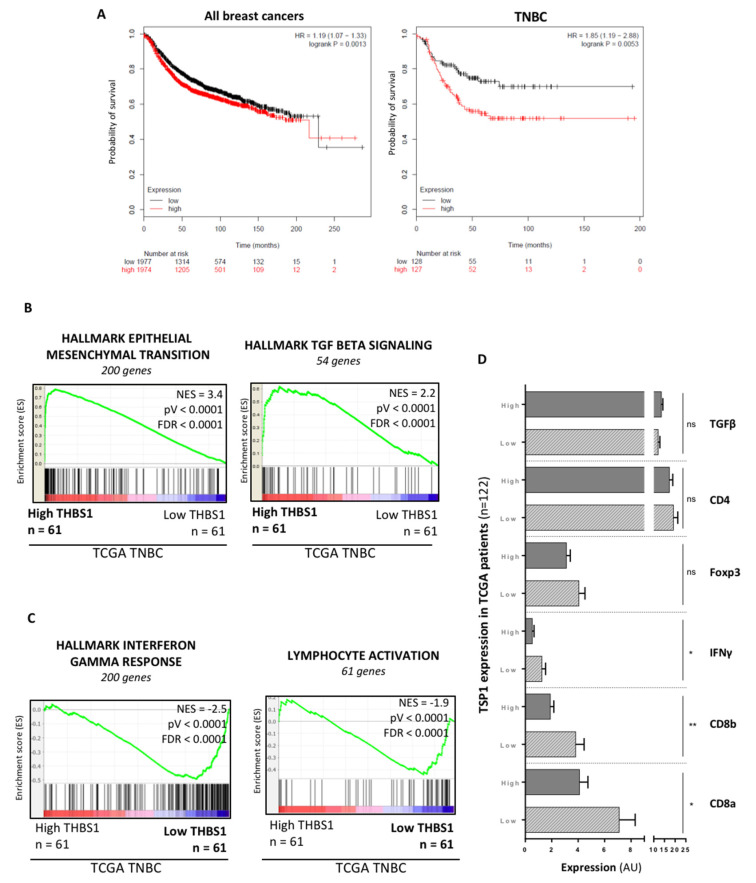
Impact of TSP1 expression on prognostic value in breast cancer. Schemes follow the same formatting. (**A**) Kaplan–Meier relapse-free survival analysis relating to TSP1 mRNA levels in 3951 breast cancer patients (left panel) and the same analysis in a cohort of 253 TNBC patients (right panel). Hazard ratios (HRs), 95% confidence intervals and log rank *p*-values are indicated in the graphs. (**B**,**C**) Gene Set Enrichment Analysis of gene signatures was performed from transcriptomes of the human TNBC TCGA database. Kolmogorov–Smirnov statistical test was performed. (**D**) mRNA expression in low and high TSP1 mRNA-expressing tumor samples from the human TNBC TCGA database. Mann–Whitney statistical test was performed. *, *p* < 0.05; **, *p* < 0.01; ns, not significant. Abbreviations: AU, Arbitrary Units; CD4, cluster of differentiation 4; CD8a, cluster of differentiation 8a; CD8b, cluster of differentiation 8b; FDR, false discovery rate; Foxp3, forkhead box protein P3; HR, Hazard Ratio; IFNγ, interferon γ; NES: normalized enrichment score; TGFβ, transforming growth factor β; TNBC, triple-negative breast cancer; TCGA, the cancer genome atlas; TSP1, Thrombospondin-1.

**Figure 2 cancers-13-04059-f002:**
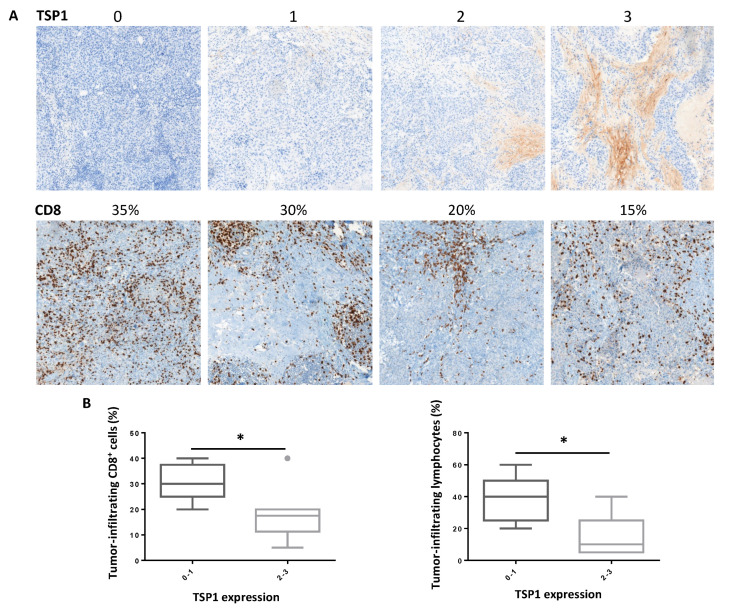
TSP1 expression and tumor-infiltrating lymphocytes in TNBC breast cancer patients. (**A**) Illustration of TSP1 immunodetection in TNBC breast cancer. The values indicate the scores associated with TSP1 expression. Examples of immunohistochemical (IHC) staining of CD8+ TILs obtained on TNBC breast cancer specimens to determine the proportion of CD8+ TILs. (**B**) Proportion of CD8+ (left panel) and total (right panel) TILs on tumor samples from 12 TNBC patients exhibiting low (score 0–1) or high (score 2–3) TSP1 expression (*: *p* < 0.05).

**Figure 3 cancers-13-04059-f003:**
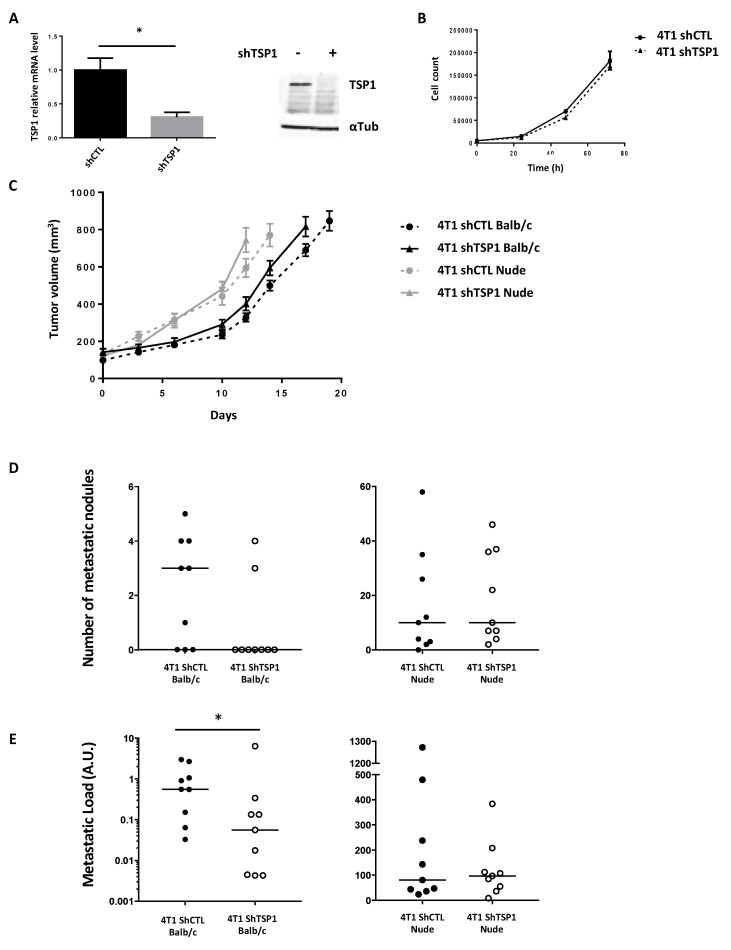
Impact of TSP1 expression in 4T1 mouse breast cancer model. (**A**) TSP1 mRNA (left panel) and protein (right panel) expression in 4T1 shCTL and 4T1 shTSP1 was evaluated by RT-qPCR and Western blot, respectively. Left panel, data are means ± SEM of *n* = 3 independent experiments. (**B**) In vitro cell proliferation of 4T1 shCTL and 4T1 shTSP1, data are means ± SEM of triplicate determination of one representative experiment of three independent experiments. (**C**) Growth of 4T1 shCTL and 4T1 shTSP1 orthotopic tumors over time in Balb-c mice (black) and nude mice (gray). (**D**,**E**) Lungs were collected from 9 mice of each group (left panels: wild-type mice; right panels: nude mice) when primary tumors reached 1000 mm^3^. Visible lung metastatic nodules on the lung surface were counted (**D**). Alternatively, RNA was extracted from lungs, and metastatic load in lungs was quantified by measurement of luciferase mRNA level normalized to cyclophilin A (**E**). Bars indicate the medians (*: *p* < 0.05).

**Figure 4 cancers-13-04059-f004:**
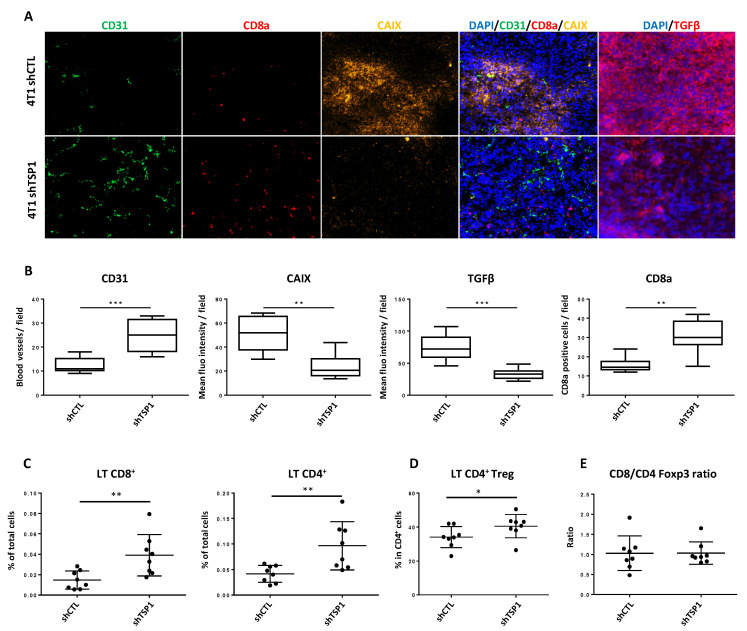
Impact of TSP1 expression on tumor microenvironment in 4T1 mouse breast cancer model. (**A**) Immunodetection of CD31, CD8α, CA-IX and active TGF-β in 4T1 shCTL and 4T1 shTSP1 tumors 11 days after cell injection in Balb/c mice. (**B**) Quantification in 4T1 shCTL and 4T1 shTSP1 tumors of blood vessels, CD8+ TILs, CA-IX and active TGF-β-positive area. Data obtained in 8 mice are represented as Tukey boxes. (**C**–**E**) Alternatively, tumors were collected at day 11 and dissociated, and the tumor cell content was analyzed by using flow cytometry (*n* = 8 mice per group). Bars are means ± SEM (*: *p* < 0.05; **: *p* < 0.01; ***: *p* < 0.001).

**Figure 5 cancers-13-04059-f005:**
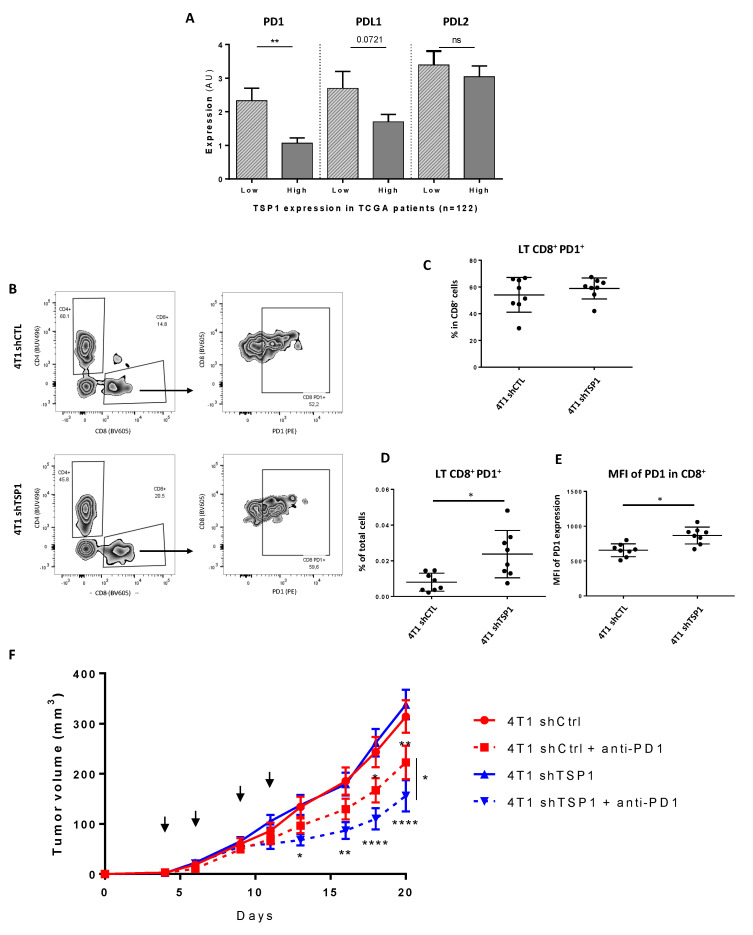
Impact of TSP1 expression on PD-1 expression and anti-PD-1 treatment in 4T1 mouse breast cancer model. (**A**) Analysis of PD-1, PD-L1 and PD-L2 expression in TCGA patients was performed depending on the median expression of TSP1 (**: *p* < 0.01). (**B**–**E**) At day 11, PD-1 expression on CD8+ TILs was quantified by flow cytometry on 4T1 shCTL and 4T1 shTSP1 tumors from Balb-c mice (*: *p* < 0.05) (**B**). Proportion of PD-1+ cells among CD8+ TILs (**C**) or total cells (**D**). Mean fluorescence intensity (MFI) of PD-1 expression on CD8+ TILs (**E**). (**F**): 1 × 10^5^ 4T1 shCTL or 4T1 shTSP1 breast carcinoma cells were orthotopically injected into the mammary fat pad of WT Balb/c mice. Mice received four injections of anti-PD-1 antibodies at days 4, 6, 9 and 11 (black arrows) at 10 mg/kg. Alternatively, mice were injected with isotype control. Tumor volumes were determined with a caliper at the indicated days. Data are means ± SEM obtained from 8 mice per group from one experiment. A two-way ANOVA with the Tukey multiple comparison test was used, and differences were statistically significant (*: *p* < 0.05; **: *p* < 0.01; ****: *p* < 0.0001).

## Data Availability

The data that support the findings of this study are available from the corresponding author upon reasonable request.

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
