# Peer review of "Thrombospondin-1 Silencing Improves Lymphocyte Infiltration in Tumors and Response to Anti-PD-1 in Triple-Negative Breast Cancer"

_cancers, 2021, doi:10.3390/cancers13164059_

Round 1

Reviewer 1 Report

The MS conducted TSPI silencing and the potential anti-PD-1 therapy in 12 cases of TNBC and few BCC

The fundamental concept seems good but requires further clarification of TSPI in clinical application. The number of cases are too small for the final conclusion.

Author Response

We agree with Referee #1, our IHC analyses on human specimens have been conducted on a low number of cases (i.e., 12 patients with TNBC). This pilot study is nevertheless reinforced by our in silico analyses on 253 TNBC patients, further demonstrating inverse correlation between anti-cancer immune response gene signatures (i.e., IFN gamma response and Lymphocyte activation) and TSP1 expression. The third paragraph of our discussion has been changed and the following sentence has been added: “Whereas our pilot study based on IHC analyses is limited due to the low number of cases (i.e., 12 patients with TNBC), the inverse correlation between anti-cancer immune response and TSP1 expression is further documented by our in silico analyses on 253 TNBC patients”.

Concerning the putative clinical application of our work, “Moreover, TSP1 could be considered as a bad prognosis biomarker in patients affected with TNBC and as a therapeutic target in combination with anti-PD-1 blocking antibody” (such as nivolumab or pembrolizumab). The latter sentence has been added in the Conclusions.

Reviewer 2 Report

This manuscript by Marcheteau, et al. showed that Thrombospondin-1 (TSP1) expression in TNBC suppressed T cell infiltration (particularly, CD8+ T cells) and therapeutic response to anti-PD-1, possibly due to its roles in tumor vascularization and TGF-β activation. Interestingly, the seemingly immunostimulatory effects from TSP1 knock-down in 4T1 TNBC cells did not impact growth of the primary tumors, yet reduced lung metastasis, in a T cell-dependent fashion, based on the results from nude mice that lack T cells. Furthermore, the authors showed that TSP1 knockdown in 4T1 cells improved anti-PD-1-induced tumor suppression, which the authors attributed to increased PD-1 expression among all TILs, although PD-1 expression on CD8+ TILs was comparable between shCTL and shTSP1 tumors.

Major concerns:

  1. It seemed that TSP1 expression among all breast cancer cases did not correlate with the overall survival. They should just present TSP1 in TNBC, as the difference of patients with TSP1high vs TSP1low TNBCs is quite impressive, with respect to their long-term survival. The discrepancy between the enriched TGF-b signaling in TSP1high TNBCs vs the no difference of TGF-b between TSP1high vs TSP1low TNBC warrant additional discussion.
  2. Although results from nude mice did suggest a role of T cells in mediating anti-metastatic effects from TSP1 KD, the specific mechanism(s) remain to be illustrated. As of now, all the presented results were largely correlative and causal relationships can't be established, including inactivation of TGF-b, enhanced infiltration of CD8 T cells, increased expression of PD-1 on TILs. Additional functional analysis of TILs should be performed. Given the indiscriminate increases of CD8 and Treg in shTSP1 tumors, it is hard to conclude that TSP1 KD led to an immunostimulatory outcome. 
  3. While CD31 expression was increased in TSP1 knockdown tumors, whether or not vasculature normalization also occurred in these tumors remains to be demonstrated. As they discussed, there was no increase of HEVs and thus how TSP1 KD drives T cell infiltration is still unknown, perhaps due to antagonism of CD47, as previously reported. 
  4. How to reconcile the no-difference of CD8/Treg in TME but an increase of CD8/Treg in DLNs, upon TSP1 knockdown?
  5. PD-1 expression on TILs alone may not be sufficient to establish the axis of TSP1 KD in tumor cells-PD-1-anti-tumor responses, in the absence or presence of anti-PD-1, esp. considering that PD-1 expression can be used as an T cell activation marker or a T cell exhaustion marker. Additional markers should be included, such as TIM-3, LAG-3, etc. 
  6. In Discussion (line360), please note the differential role of Foxp3 between mouse and human. 
  7. It remains unknown how the authors plan to target TSP-1 in tumor cells. 

Minor point

Please check spelling and unify abbreviations used in this manuscript. 

Author Response

Please find enlosed our Responses to referee#2's comments.

Reviewer 3 Report

The manuscript entitled “ Thrombospondin-1 silencing improves lymphocyte infiltration in tumors and response to anti-PD-1 in triple negative breast cancer” described studies examining the effect of the anti-angiogenic Thrombospondin-1 (TSP1) in TNBC. They found that TSP1 knockdown reduced TGF-β activation and enhanced the content of TILs and decreased lung metastasis in syngeneic Balb/c immunocompetent mice. In addition, they also found TSP1 knockdown increased CD8+ TILs & PD-1 expression and sensitized 4T1 tumors to anti-PD-1 therapy. In their conclusion, the combination of targeting TSP1 and immune checkpoint inhibitors can be a new strategy in treatment of TNBC.

The basic question is interesting and the data are consistent with most of the notion. And the paper is well written and discussed. I suggest this article can be accepted and published

Author Response

We thank referee #3 for his/her positive comment.

Round 2

Reviewer 1 Report

I think that it is improved with revision and it is another concept how we explore a potential therapeutic target in TNBC which is one of my interest.

But I am still questioning about the hypothesis based on known-down model and how clinically this can be  applicable.

However, it is acceptable for publication.

Reviewer 2 Report

Please read recent relevant articles on TSP-1 in BC. 

2019 Cancers "Exosomal Thrombospondin-1 Disrupts the Integrity of Endothelial Intercellular Junctions to Facilitate Breast Cancer Cell Metastasis" and a recent review article on TSP-1 role in T cells.